# Genomic Characterization and Environmental Distribution of a Thermophilic Anaerobe *Dissulfurirhabdus thermomarina* SH388^T^ Involved in Disproportionation of Sulfur Compounds in Shallow Sea Hydrothermal Vents

**DOI:** 10.3390/microorganisms8081132

**Published:** 2020-07-27

**Authors:** Maxime Allioux, Stéven Yvenou, Galina Slobodkina, Alexander Slobodkin, Zongze Shao, Mohamed Jebbar, Karine Alain

**Affiliations:** 1Univ Brest, CNRS, IFREMER, LIA1211, Laboratoire de Microbiologie des Environnements Extrêmes LM2E, IUEM, Rue Dumont d’Urville, F-29280 Plouzané, France; Maxime.Allioux@univ-brest.fr (M.A.); steven.yvenou@gmail.com (S.Y.); Mohamed.Jebbar@univ-brest.fr (M.J.); 2Winogradsky Institute of Microbiology, Research Center of Biotechnology of the Russian Academy of Sciences, 117312 Moscow, Russia; gslobodkina@mail.ru (G.S.); aslobodkin@hotmail.com (A.S.); 3Key Laboratory of Marine Genetic Resources, Third Institute of Oceanography, Ministry of Natural Resources, Xiamen 361005, China; shaozongze@tio.org.cn

**Keywords:** genome annotation, *Dissulfurirhabdus*, shallow sea hydrothermal vents, inorganic sulfur compound disproportionation

## Abstract

Marine hydrothermal systems are characterized by a pronounced biogeochemical sulfur cycle with the participation of sulfur-oxidizing, sulfate-reducing and sulfur-disproportionating microorganisms. The diversity and metabolism of sulfur disproportionators are studied to a much lesser extent compared with other microbial groups. *Dissulfurirhabdus thermomarina* SH388^T^ is an anaerobic thermophilic bacterium isolated from a shallow sea hydrothermal vent. *D. thermomarina* is an obligate chemolithoautotroph able to grow by the disproportionation of sulfite and elemental sulfur. Here, we present the results of the sequencing and analysis of the high-quality draft genome of strain SH388^T^. The genome consists of a one circular chromosome of 2,461,642 base pairs, has a G + C content of 71.1 mol% and 2267 protein-coding sequences. The genome analysis revealed a complete set of genes essential to CO_2_ fixation via the reductive acetyl-CoA (Wood-Ljungdahl) pathway and gluconeogenesis. The genome of *D. thermomarina* encodes a complete set of genes necessary for the dissimilatory reduction of sulfates, which are probably involved in the disproportionation of sulfur. Data on the occurrences of *Dissulfurirhabdus* 16S rRNA gene sequences in gene libraries and metagenome datasets showed the worldwide distribution of the members of this genus. This study expands our knowledge of the microbial contribution into carbon and sulfur cycles in the marine hydrothermal environments.

## 1. Introduction

Anaerobic microorganisms are involved in biogeochemical cycles and are vital for global ecosystem maintenance, including marine hydrothermal vents. Shallow hydrothermal vents, like deep-sea hydrothermal vents, are areas characterized by the discharge of hot, anoxic, mineral-loaded, and reduced compound-rich fluid into the cold and oxygenated water of the ocean floor [1,2]. Sulfur is a ubiquitous element in the hydrothermal environment and is very important for energy production. It is found in various oxidation states in the mineral structures forming chimneys, in the fluid emitted from the chimneys, especially as hydrogen sulfide (H_2_S) and in the surrounding sea-water as sulfate. The sulfur-oxidizing and sulfur/sulfate-reducing microbial taxa of these habitats are well known [3]. However, it is only very recently that sulfur-disproportionating species from this ecosystem have been reported there, even though the physico-chemical conditions of this habitat are obviously favorable to this reaction. To date, five sulfur-disproportionating species, all thermophilic, have been isolated from marine hydrothermal environments. The bacteria *Thermosulfuriphilus ammonigenes*, *Dissulfuribacter thermophilus* and *Thermosulfurimonas dismutans* were isolated from deep-sea hydrothermal vents [4,5,6], and the species *Thermosulfurimonas marina* and *Dissulfurirhabdus thermomarina* from shallow hydrothermal systems [7,8].

Disproportionation, also called dismutation, corresponds to a chemical or biological reaction where the same mineral or organic compound serves as an electron donor and as an electron acceptor. The microbially-mediated disproportionation of inorganic sulfur compounds was first described in 1987 [9,10]. Diverse inorganic sulfur compounds can be disproportionated: generally, the most studied forms for sulfur disproportionation are elemental sulfur (S^0^), thiosulfate (S_2_O_3_^2−^) and sulfite (SO_3_^2−^), which can be both oxidized to sulfate (SO_4_^2−^) and reduced to sulfide (HS^−^) [11]. Under standard conditions, disproportionation reactions have a mainly low energy yield or can even be endergonic for elemental sulfur, according to thermodynamics, but can shift to be more energetic depending on the physico-chemical conditions of the natural environments and/or the presence of possible sulfide scavenging species. Few species have also been reported to disproportionate sulfur compounds solely for energy production and not for growth [11], as a kind of maintenance process, which could increase survival under limiting conditions. Most of the disproportionating microorganisms are known to use alternative energetically more favorable processes, such as sulfate-reduction or dissimilatory nitrate reduction to ammonium [12]. Interestingly, the disproportionation of elemental sulfur could date back up to 3.5 Ga and could be one of the earliest modes of microbial metabolism [13,14], but this hypothesis still remains highly controversial [15].

Sulfur compounds disproportionators originate from a large panel of environments such as marine sediments, freshwater sediments, anaerobic digestors, terrestrial, shallow and deep-sea hydrothermal vents, and acidic and alkaline lakes [12]. This process has been extensively studied in marine sediments but not in other environments [12,16,17]. In the current state of knowledge, and with recent discoveries, sulfur compounds disproportionating microorganisms appear to be phylogenetically diverse, particularly in the bacterial domain. From the literature, we could elaborate a list of 42 bacterial species in total, known to be able to disproportionate inorganic sulfur compounds, and being independent of the sulfur oxygenase reductase (SOR) enzyme [11,12]. The microorganisms known so far to be capable of disproportionating sulfur compounds under anaerobic conditions belong to *Thermodesulfobacteria*, *Firmicutes*, *Deltaproteobacteria* and *Gammaproteobacteria* [12].

To date, the metabolic pathways of sulfur-compound disproportionation and the importance of this process remain poorly documented, notably due to the absence of specific genomic markers.

Pathways of the disproportionation of sulfur are unknown and different pathways are very likely to exist. Some hypotheses have been proposed, such as the use of the complete or partial dissimilatory sulfate reduction pathway (adenylylsulfate reductase, heterodisulfide reductase, dissimilatory sulfite reductase), or the involvement of rhodanese-like sulfurtransferase or molybdopterins [18,19,20]. As suggested by Ward et al. (2020) [21], a truncated AprB protein may also be involved in this process but this modified protein does not appear to be common in all sulfur disproportionators. Finally, Mardanov et al. (2016) [18] showed that direct cellular contact with sulfur is not required. As suggested in Florentino et al. (2019) [19], certain molecular strategies could be involved in the assimilation of sulfur in cells, such as the formation of sulfur nanoparticles that can penetrate membranes, the nucleophilic attack of sulfur by sulfides that could generate polysulfides, used as a source of energy, or strategies involving flagella or pili.

*Dissulfurirhabdus thermomarina* SH388^T^ is an anaerobic, thermophilic, chemolithoautotrophic bacterium isolated from a shallow submarine hydrothermal vent, located off the Kuril Islands (44°29.469′ N 146°06247′ E), in the Sea of Okhotsk, at a water depth of 12 m [7]. It is the first strict anaerobic thermophilic species disproportionating inorganic sulfur compounds which was isolated from a shallow sea habitat. Based on its 16S rRNA gene sequence, it belongs to the class *Deltaproteobacteria* and is closely related to *Dissulfuribacter thermophilus* S69^T^ and *Dissulfurimicrobium hydrothermale* SH68^T^. Physiological experiments demonstrated that *D. thermomarina* strain SH388^T^ grows chemolithoautotrophically with bicarbonate/CO_2_ as a carbon source, either by the respiration of sulfite coupled to the oxidation of dihydrogen, or by the disproportionation of sulfite or elemental sulfur [7]. However, it does not grow by thiosulfate disproportionation. In this work, we analyzed the genome and the geographical distribution of *D. thermomarina* SH388^T^. A very recent study also looked at the genome of *D. thermomarina* and in particular at its phylogenetic positioning [21]. In this survey, we did not rely on the assembly of this genome already available in the RefSeq and GenBank databases (accession number ASM1049943v1) used in Ward et al.’s (2020) [21] study, but we sequenced this genome de novo and made our own assembly, as detailed below, with more in-depth genome assembly and annotation strategies. Based on our genome assembly, we have highlighted the general metabolic pathways of this strain, and focused in particular on the energy production pathways involving sulfur inorganic compounds. Genome sequence availability and annotation will promote a better understanding of the genomic traits of a sulfur compound disproportionating bacteria, the metabolic features related to the adaptations to the ecosystem, and will be useful for future sulfur cycle studies.

## 2. Materials and Methods

### 2.1. Genome Sequencing and Assembly

For genomic DNA extraction, the strain was cultivated anaerobically at 50 °C, with H_2_ as an electron donor, sulfite (5 mM) as a terminal electron acceptor and CO_2_/HCO_3_^−^ as the sole carbon source. Cells were harvested in the late exponential phase of growth. Genomic DNA was extracted using a FastDNA™ Spin Kit (MP Biomedicals, Irvine, CA, USA) according to the manufacturer’s instructions. The genome sequence of strain SH388^T^ was determined by the company Molecular Research (MrDNA, Shallowater, TX, USA) using the Illumina MiSeq technology (2 × 150 bp paired-reads, MicroV2 chemistry). Libraries’ constructions and quality controls were performed by both sequencing facilities and verified with FastQC (v0.11.8—https://www.bioinformatics.babraham.ac.uk/projects/fastqc/). Genome was assembled into contigs by using the Unicycler pipeline for the de novo assembly (version: 0.4.8-beta—https://github.com/rrwick/Unicycler), and its dependencies (spades.py v3.14.0; makeblastdb v2.9.0+; tblastn v2.9.0+; bowtie2-build v2.3.5.1; bowtie2 v2.3.5.1; samtools v1.10; java v11.0.1; pilon v1.23; bcftools v 1.10.2) [22]. Genome assembly statistics were obtained with Quast (v5.0.2; https://github.com/ablab/quast) and used to compare the different assemblies. Genome assembly visualization was plotted with Bandage (v0.8.1—http://rrwick.github.io/Bandage/) in order to detect potential plasmids from obtained contigs and afterwards checked with plasmidVerify python script (https://github.com/ablab/plasmidVerify) [23,24]. Genome completeness and potential contamination were controlled with CheckM (v1.1.2—https://ecogenomics.github.io/CheckM/), and whole genome average coverage was calculated using BBMap (v38.70—BBMap—Bushnell B.—sourceforge.net/projects/bbmap/).

### 2.2. Genome Annotation

Genome was analyzed and annotated with the online version of the RAST software (v2.0—http://rast.theseed.org/FIG/rast.cgi), the fast annotation software Prokka (v1.14.6—https://github.com/tseemann/prokka), Dfast (v1.2.5—https://github.com/nigyta/dfast_core), the MicroScope Microbial Genome Annotation and Analysis Platform (MaGe) (https://mage.genoscope.cns.fr/microscope/home/index.php), using the Kyoto Encyclopedia of Genes and Genomes (KEGG) and BioCyc databases, and the NCBI prokaryotic genome annotation pipeline (PGAP) (2020-03-30.build4489—https://github.com/ncbi/pgap) with default parameters and databases for all of the five software/pipelines [25,26,27,28,29]. The functional annotation of predicted coding DNA sequences (CDSs) was further blasted with NCBI (v2.10.0+), and UniProtKB database (release 2020_04). Hydrogenase classification was checked using the HydDB webtool (https://services.birc.au.dk/hyddb/) [30].

### 2.3. Clustered Regularly Interspaced Short Palindromic Repeats (CRISPRs) and Genomic Islands

Identification and classification of the CRISPR–Cas systems were performed by using the CRISPRCas Finder webserver, with default parameters (https://crisprcas.i2bc.paris-saclay.fr/) [31]. The prediction of laterally transferred gene clusters (genomic islands) was performed with the IslandViewer4 webserver (http://www.pathogenomics.sfu.ca/islandviewer/) based on an EMBL file generated by Dfast [32].

### 2.4. Geographical Distribution

The geographical distribution of *D. thermomarina* was studied at species and genus level within the 16S rRNA gene sequences available in the databases and in the public metagenomes deposited at the GBIF (Global Biodiversity Information Facility) facility (https://www.gbif.org/) and in the NCBI database.

### 2.5. Taxonomical Analyses and Comparative Genomics

To study the taxonomic position of the strain, we used GTDB-Tk (v1.1.1—https://github.com/Ecogenomics/GTDBTk) to place the genome on a tree made of concatenated reference proteins, we compared by blast the 16S rRNA CDS obtained from genomic assembly to the sequences in NCBI (v2.10.0+) and performed a tetra correlation comparison search with the JSpecies webserver against its own database (http://jspecies.ribohost.com/jspeciesws/).

The genome of *D. thermomarina* was compared by subtractive comparative genomics to the genomes of the hydrothermal bacteria *Thermosulfurimonas marina* (ASM1231758v1), *Thermosulfuriphilus ammonigenes* (ASM1120745v1), *Dissulfuribacter thermophilus* (ASM168733v1), and *Thermosulfurimonas dismutans* (ASM165258v1) to identify potential genetic markers of DNRA, and of thiosulfate disproportionation, two physiological properties absent in *D. thermomarina*, explored by excluding *D. thermomarina*’s genome. These genomes were compared by using the MaGE platform Pan-genome Analysis tool (https://mage.genoscope.cns.fr/microscope/home/index.php), based on the clustering algorithm SiLiX (http://lbbe.univ-lyon1.fr/-SiLiX-.html) which clustered genomic CDSs by 50% amino acid identity and 80% amino acid alignment coverage, with permissive parameters. Resulting CDSs were blasted on the UniprotKB database and hypothetical protein CDSs were analyzed with InterProScan webserver (https://www.ebi.ac.uk/interpro/) for functional predictions.

Finally, to evaluate the hypothesis of Ward et al. (2020) [21] suggesting that the tail truncation of the AprB protein could be a molecular marker of the disproportionation capacity of sulfur, we extracted the CDS encoding the AprB protein (based on Prokka annotation) from the genomes of characterized sulfur disproportioners or sulfate reducers: *Thermosulfurimonas marina*, *Thermosulfuriphilus ammonigenes*, *Thermosulfurimonas dismutans*, *Dissulfuribacter thermophilus, Thermodesulfatator atlanticus, Thermodesulfatator autotrophicus* and *Thermodesulfatator indicus,* in addition to that of *D. thermomarina* [4,5,6,7,8,33,34,35]. The AprB putative protein sequences were then aligned and their length were calculated.

## 3. Results and Discussions

### 3.1. General Genome Properties and Genomic Islands

The complete genome sequence of *Dissulfurirhabdus thermomarina* strain SH388T was deposited in GenBank databases under the accession number JAATWC000000000. The strain is available in the DSMZ culture collection under the accession number DSM 100025T and in All-Russian Collection of Microorganisms (VKM) under the accession number VKM B-2960T. The *D. thermomarina* SH388^T^ genome sequence consisted of 36 contigs including two contigs of less than 200 bp with an overall size of 2,461,642 bp and a G + C content of 71.1 mol% (Figure 1). It is interesting to note that we found a higher G + C content than in Slobodkina et al. (2016) [7]. In this previous study, the DNA G + C content value was determined from the melting point with DNA of *Escherichia coli* K-12 as a reference. This method is more biased than determining the percentage of G + C directly from the genomic sequence. We therefore proposed to amend the description of the species *Dissulfurirhabdus thermomarina* in that respect.

The longest contig was 713,503 bp, the *N_50_* was 240,491 bp and the *L_50_* was 3. The quality of this assembly was superior to that deposited previously in databases under the accession number ASM1049943v1 (Illumina HiSeq sequencing; draft genome of 2,569,312 bp in length, for 386 contigs with a *N_50_* of 14,884 bp and a *L_50_* of 54). The genome could be made up of one circular chromosome; indeed, no plasmids were detected when the genome was plotted with Bandage and by applying the plasmidVerify script to all contigs. CheckM estimated the genome to be 98.1922% complete based on the presence of default single-copy marker genes (four markers were missing) and without any hypothetical contamination. The average genome coverage was extremely high, around 1384.362× according to raw pair read sequences extracted from MiSeq sequencing data (Table 1).

Annotation with PGAP resulted in the prediction of 2321 genomic objects, among which 2267 were protein-coding sequences. The strain had a relatively streamlined genome with coding sequences covering approximately 90.8% of the entire genome. However, slightly different results were obtained with other annotation software: 2407 CDSs were found with RAST (1898/2407 were not integrated to subsystem categories), 2221 CDSs with Prokka, 2250 CDSs with Dfast and 2280 CDSs with MaGe annotation. The genome also contained one operon of 5S-16S-23S rRNA genes. We detected 47 tRNA with MaGe and PGAP which use the tRNA scan-SE RNA finder (http://lowelab.ucsc.edu/tRNAscan-SE/index.html), while 58 tRNA were detected with Prokka and Dfast based on ARAGORN RNA finder (http://130.235.244.92/ARAGORN/). However, the tRNA found in all cases corresponded to the 20 standard amino acids and selenocysteine. These results differ somewhat from those of Ward et al. (2020) [21] whom reported 2791 coding sequences and 53 RNAs, on a lower quality assembly of the genome.

Most of the CDSs obtained from the MaGe annotation pipeline (81.80%, 1865/2280 CDSs) could be assigned to at least one cluster of orthologous groups (COGs). The major predicted COG categories (encompassing more than 2% of the CDSs) were related to energy production and conversion (C) (8.2%), signal transduction mechanisms (T) (7.6%), translation-ribosomal structure-biogenesis (J) (7.1%), cell wall/membrane/envelope biogenesis (M) (6.9%), amino acid transport and metabolism (E) (6.3%), inorganic ion transport and metabolism (P) (5.2%), posttranslational modification-protein turnover-chaperones (O) (4.9%), coenzyme transport and metabolism (H) (4.6%), cell motility (N) (4.3%), replication–recombination–repair (L) (4.2%), carbohydrate transport and metabolism (G) (3.9%), transcription (K) (3.8%), intracellular trafficking-secretion-vesicular transport (U) (3.4%), lipid transport and metabolism (I) (2.4%), nucleotide transport and metabolism (F) (2.3%) and secondary metabolites biosynthesis, transport and catabolism (Q) (2.1%).

The gene locus tags associated to the genome assembly annotation given in GenBank and RefSeq (ASM1297923v1) are reported in the Appendix A.

Furthermore, four potential CRISPR loci were found using the CRISPRCasFinder server. These loci consisted of three putative CRISPR systems containing one spacer and a fourth one confirmed as a CRISPR locus, which was 466 bp long and included six spacers and repeats of 35 bp (GAAGGAATTGACCTGATTACTGAAGGGATTACGAC), but without cas genes.

Two regions of genomic plasticity were identified with the IslandViewer4 webserver, by using the genome of the closest database relative *Desulfuromonas* sp. DDH964 as a reference genome. These two genomic islands (GI) had a total length of 41.8 kb. The first GI was composed of genes involved in carbohydrate biosynthesis, degradation and transport-related CDSs (mostly mannose), as opposed to the second GI, composed mostly of hypothetical proteins CDSs (Appendix A).

Interestingly, *D. thermomarina* seems to be quite unique among all the cultivated representatives of the *Bacteria* domain; based on 16S rRNA gene sequence identity, Average Nucleotide Identity (ANI) and phylogenetic placement based on concatenated references proteins (GTDB-Tk), we did not find any close relatives except *Dissulfuribacter thermophilus* strain S69^T^ and *Dissulfurimicrobium hydrothermale* strain Sh68^T^, two *Deltaproteobacteria*, as already demonstrated by Slobodkina et al. (2016) [7]. *Dissulfurirhabdus thermomarina* SH388^T^ was distantly related to these two closest cultured species, with its 16S rRNA gene sequence displaying only 91.6% and 90.4% gene sequence similarity with the 16S rRNA gene sequences of the *Dissulfuribacter thermophilus* strain S69^T^ and *Dissulfurimicrobium hydrothermale* strain Sh68^T^, respectively. GTDB-Tk classified *D. thermomarina* within the *Dissulfuribacterales* order, but has not associated it with any family or genus. The tetranucleotide signatures search showed strong similarities with species belonging to the *Gammaproteobacteria*, the *Actinobacteria* and the *Firmicutes* (Z-score > 0.9). On the basis of all these results, the proposal by Ward et al. (2020) [21] to assign *D. thermomarina* to a new family appears justified.

### 3.2. Central Carbon Metabolism

*D. thermomarina* SH388^T^ is capable of growing autotrophically from CO_2_/HCO_3_^−^ [7]. We found a complete Wood–Ljungdahl pathway (reductive acetyl-CoA pathway) for carbon dioxide fixation and the generation of acetyl-CoA by integrating the annotations of PGAP, Biocyc, KEGG and Prokka (Appendix A). Based on the enzymes detected with MicroCyc, the strain appears also to have a complete glycolysis (Embden–Meyerhof), gluconeogenesis and pentose phosphate pathways. *D. thermomarina* seems also to possess some CDSs associated to the formate dehydrogenase. However, the capacity of *D. thermomarina* to oxidize formate into CO_2_ has not been demonstrated experimentally, and Slobodkina et al. (2016) [7] demonstrated that formate does not stimulate the growth of *D. thermomarina*. Based on the KEGG and Biocyc databases, the tricarboxylic acid (TCA) cycle appears incomplete. Most probably, it serves for the formation of the necessary biosynthetic intermediates, in particular oxaloacetate, succinyl-CoA and 2-oxoglutarate. A key enzyme for the reverse TCA pathway, ATP-citrate lyase, is missing. According to KEGG and MicroCyc, *D. thermomarina* also appears to have a complete glycogen degradation pathway allowing the degradation of glycogen to G6P, or the reverse reaction. We also found a pyruvate fermentation pathway, oxidizing pyruvate to acetyl-CoA but Slobodkina et al. (2016) [7] showed experimentally that *D. thermomarina* does not ferment pyruvate. The addition of pyruvate in the medium could prevent the conversion of acetyl-CoA to pyruvate by the pyruvate synthase, and pyruvate could theoretically serve as a direct carbon substrate for gluconeogenesis. Nevertheless, according to MicroCyc, no other fermentative pathways were found, which is congruent with the autotrophic nature of the strain. In its natural habitat, this strain therefore probably develops from the CO_2_ emitted in the hydrothermal fluid, and in other ecosystems, from the CO_2_ produced by the microbial metabolism or abiotically.

### 3.3. Hydrogen Metabolism

*D. thermomarina* is capable of using hydrogen as an energy source [7]. Prokka, PGAP and MaGe annotations detected several hydrogenase-related proteins: maturation factors, hydrogenase formation chaperone, hydrogenases subunits and hydrogenase expression proteins (Appendix A). *D. thermomarina* appears to have a complete gene cluster encoding a membrane-bound [NiFe]-hydrogenase, belonging to the Group 1c [NiFe]-hydrogenase according to the HydDB classifier. Small and large subunit CDSs, as well as maturation factors, were found, but we were not been able to clearly distinguish the four hydrogenase subunits HybO, HybA, HybB and HybC. This hydrogenase is likely to be involved in the anaerobic H_2_-uptake, for the hydrogenotrophic respiration with sulfite or SO_2_ gas as terminal electron acceptors. As hydrothermal fluids are generally charged with dihydrogen (with particularly high concentrations at ultramafic sites), this highly energetic source feeds the autotrophic microorganisms inhabiting these unique habitats, such as *D. thermomarina*. In other anoxic habitats, microbial fermentations produce H_2_, as well as a number of abiotic reactions.

### 3.4. Nitrogen Metabolism

Species isolated from hydrothermal vents such as *Thermosulfurimonas dismutans*, *Thermosulfurimonas marina*, *Thermosulfuriphilus ammonigenes* and *Dissulfuribacter thermophilus* demonstrated the ability to use nitrate as an electron acceptor by performing DNRA metabolism [36]. We did not find any strong evidence for an energetic metabolism based on nitrogen compounds in *D. thermomarina*, which is congruent with the culture/physiology results [7]. Nevertheless, the genome contains a hydroxylamine oxidoreductase (EC: 1.7.2.6) and a hydroxylamine reductase (EC: 1.7.99.1) as evidenced by PGAP and Prokka annotations. An ammonia transporter and two P-II family nitrogen regulator CDSs were also found with Prokka. Nitrogen uptake pathways may not be canonical as no complete pathways were found, with the exception of one glutamine synthetase (EC: 6.3.1.2). From these results, *D. thermomarina* seems unlikely to participate to the global environmental nitrogen cycle.

### 3.5. Sulfur Metabolism

As has been shown for many bacteria, listed in the review by Slobodkin and Slobodkina (2019) [12], a complete sulfate reduction pathway was found in the genome of *D. thermomarina,* despite the fact that physiological experiments conducted in vitro showed that this strain does not grow from sulfate reduction [7]. Based on Prokka, Dfast and PGAP annotations, a complete dissimilatory sulfate reduction pathway was found, but no assimilatory sulfate reduction path (Appendix A). We found two CDSs associated to sulfate adenylyltransferases (*sat*) (EC: 2.7.7.4) displaying 29.25% identity with each other, both subunits alpha and beta of adenylyl-sulfate reductase (*apr*A, *apr*B) (EC: 1.8.99.2), a manganese-dependent inorganic pyrophosphatase, and subunits alpha, beta and gamma of dissimilatory sulfite reductase (DsrA, DsrB, and DsrC) (EC 1.8.99.5). A dissimilatory sulfite reductase D (DsrD) CDS was also found, but only with Prokka. CDSs corresponding to a complete DsrMKJOP complex were only found with RAST annotation, and they were confirmed to be related to menaquinol oxidoreductases by comparison to the UniProtKB database. A complete APS reductase-associated electron transfer complex (QmoABC) was found with PGAP and UniprotKB, if we refer to their homology with the QmoABC CDSs of *D. thermophilus*. Based on the complete pathways present in its genome, *D. thermomarina* would have the genetic potential to couple H_2_ oxidation to sulfate reduction and should be able to grow through this metabolism; however, physiological results did not validate this hypothesis. Since *D. thermomarina* can reduce sulfite, the enzymes involved in the first step of the dissimilatory reduction of sulfate to sulfite, that are present in the genome, could have been good candidates for catalyzing the oxidation of sulfite to sulfate. However, these enzymes are not known to catalyze the reverse reaction of sulfate reduction to sulfite. In order to search for the genes involved in the disproportionation of sulfur, the genes known to be involved in the oxidation or reduction of inorganic sulfur compounds were searched for. None of the marker genes related to sulfur oxidation based on the genes cited in the recent review by Wasmund et al. (2017) [16] (e.g., sulfide:quinone oxidoreductase, Sox associated proteins, etc.) were found with any of the annotation software used. Genes encoding for sulfur oxygenase reductases (SOR), an enzyme performing elemental sulfur disproportionation under aerobic to microaerophilic conditions that had been found in the genome of the geothermal bacterium *Aquifex aeolicus* [37], was also searched for, but was not detected in *D. thermomarina*’s genome. As suggested previously, one can assume that all these CDSs attributed to dissimilatory sulfate reduction might be involved in inorganic sulfur disproportionation, through a currently undescribed process, with very likely an involvement of the adenylylsulfate reductase and the sulfate adenylyltransferase [11,12]. In addition, we found several CDSs without a clear determined function related to thiosulfate, tetrathionate and polysulfide molecules with PGAP and Prokka (polysulfide, tetrathionate and thiosulfate sulfurtransferase, reductase and dehydrogenase). These enigmatic CDSs might be as well be somehow related to sulfur compound disproportionation. Moreover, a TorD-like chaperon protein, four molybdopterin oxidoreductases and two rhodanese-like domain-containing proteins were found, as found in the genome of the alkaliphilic deltaproteobacterium *Desulfurivibrio alkaliphilus*, and hypothetically correlated to the oxidation of sulfides to sulfur by Thorup et al. (2017) [19]. Furthermore, thiosulfate cannot be disproportionated by *D. thermomarina* but one putative thiosulfate sulfurtransferase was identified in the genome by Prokka which shares 35% amino acids sequence identity with unreviewed proteins on UniprotKB database. However, considering the fact that the strain is phylogenetically distant from any cultivated representatives and relatively isolated within the bacterial domain, it is difficult to compare its CDSs to pertinent references.

These results highlight then the involvement of *D. thermomarina* into the sulfur cycle, (i) in particular in the reduction of sulfites, and (ii) somehow, still not well understood, in the disproportionation of sulfur, and (iii) finally in the sulfite disproportionation, possibly through the reverse dissimilatory sulfate reduction pathway.

### 3.6. Comparative Genomics

*D. thermomarina* was compared by subtractive comparative genomics to other genomes of hydrothermal bacteria with slightly different metabolic properties to identify in particular the potential genetic markers of DNRA and thiosulfate disproportionation. With this approach, 47 genes present in the genome of the four bacteria performing DNRA and thiosulfate dismutation and absent in the genome of *D. thermomarina* were identified (Appendix A). These CDSs are linked to the reactions of the nitrogen cycle including a periplasmic Nap-type nitrate reductase and a [FeMo]-nitrogenase (NifDKH). With regard to the disproportionation of thiosulfate, no CDS candidates were identified using this approach, with the exception of tetrathionate reductase subunit A for which the functional assignment is uncertain. A large number of hypothetical proteins were present in the subtracted gene pool, but without clear involvement in DNRA or thiosulfate dismutation reactions by InterProScan search. The thiosulfate disproportionation pathway and the DNRA route will therefore need to be studied using functional approaches in order to be deciphered.

In addition, CDS coding for AprB proteins was also analyzed for all these bacteria and we found, as in the results of Ward et al. (2020) [21], truncated proteins only in *Desulfovibrionales* and *Thermodesulfobacteriales*. However, by studying the length of CDS coding for AprB from different members of *Thermodesulfobacteriales* and comparing them to the metabolic properties of these strains, we were unable to correlate the length of these sequences with the ability or inability to disproportionate inorganic sulfur compounds. Indeed, the CDS coding for the AprB of the sulfur compounds disproportionating bacteria *Thermosulfurimonas dismutans*, *Thermosulfurimonas marina*, *Thermosulfuriphilus ammonigenes* are composed of 154 amino acids. On the other hand, the CDS coding for the AprB proteins of the sulfate-reducing bacteria *Thermodesulfatator atlanticus*, *Thermodesulfatator autotrophicus* and *Thermodesulfatator indicus* are composed of 150, 151 and 150 amino acids, respectively. Finally, the CDS coding for the AprB proteins of the sulfur disproportionators *Dissulfurirhabdus thermomarina* and *Dissulfuribacter thermophilus* are 148 amino acids long. More models are required to evaluate the hypothesis that an AprB gene truncation is associated to sulfur disproportionation, but seems unlikely to be, at least at the *Bacteria* domain scale.

### 3.7. Geographical and Environmental Distribution

The GBIF application (https://www.gbif.org/species/) enabled us to find the occurrences of the 16S rRNA gene sequences of the thermophilic sulfur disproportionators in gene libraries and metagenome datasets obtained from samples collected worldwide. While these data do not provide a comprehensive quantitative assessment, they allow to evaluate the geographical distribution of these bacteria. The analysis showed that among thermophilic sulfur-disproportionating bacteria, representatives of *Deltaproteobacteria* are more widespread than the representatives of *Thermodesulfobacteria* (Table 2).

The survey also revealed that the habitats of *D. thermomarina* are not limited to shallow sea hydrothermal vents, but also include marine coastal sediments, marine benthic sediments, ocean water column, pond soils, salt marshes or lagoon sediments contaminated with PAHs (https://www.gbif.org/species/9334679) (Figure 2A). In addition, 200 occurrences of the genus *Dissulfurirhabdus* of which 114 georeferenced were found primarily in different parts of the world ocean (https://www.gbif.org/species/9334679) (Figure 2B). Since no studied thermophilic sulfur disproportionators are known to form endospores or dormant cells, we can assume that they are in active metabolic state.

These results suggest that the diversity of the genus *Dissulfurirhabdus* is far from being explored, that this genus is distributed worldwide and could be involved in the global sulfur cycle in specific anoxic niches.

## 4. Conclusions

*D. thermomarina* belongs to a little studied deeply branched phylogenetic group. The whole-genome annotation indicates its involvement in the sulfur cycle in shallow sea hydrothermal vents. The results found were generally supporting the main metabolic features demonstrated experimentally [7] and strengthen and complement the annotation performed by Ward et al. (2020) [21]. One interesting feature is that this species could reduce sulfite but not sulfate, even if the potential genomic resources are present. This genome analysis will potentially lead to a better understanding of inorganic sulfur compound disproportionation and sulfite reduction processes. In the future, functional approaches will have to be used to decipher the pathways of inorganic sulfur compounds disproportionation, and validate the functional hypotheses derived from genomic data. It is important to know the taxa that carry out the dismutation of inorganic sulfur compounds in natural habitats as this process is not generally considered as such in global geochemical budgets. Indeed, it is crucial to determine what is the share of sulfur dismutation in the fluxes of sulfur species in habitats compared to those of sulfur-oxidation and sulfate-reduction, as dismutation is confused with these pathways in global budgets, since it leads to the production of sulfates and sulfides. Sulfur-disproportionating taxa do not necessarily have the same ecophysiological properties as sulfur-oxidizers and sulfate-reducers, and this could have a significant impact on our understanding of the biotic cycle of sulfur in natural environments.

## Figures and Tables

**Figure 1 microorganisms-08-01132-f001:**
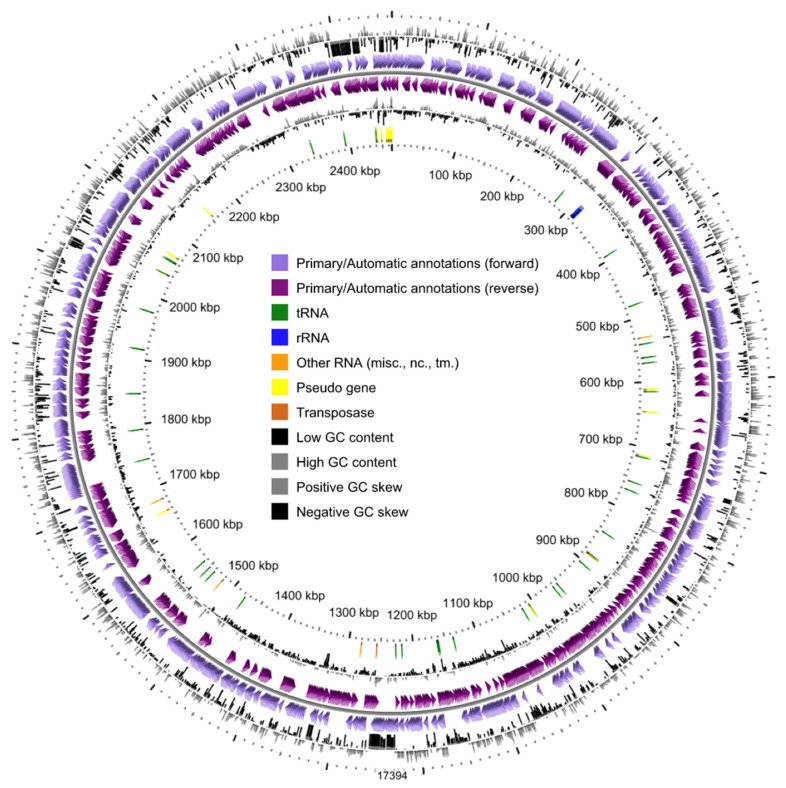
Circular mapping of the genome of *Dissulfurirhabdus thermomarina* SH388^T^ from the circular genome viewer of the MaGe platform. GC content: guanine-cytosine content (mol%).

**Figure 2 microorganisms-08-01132-f002:**
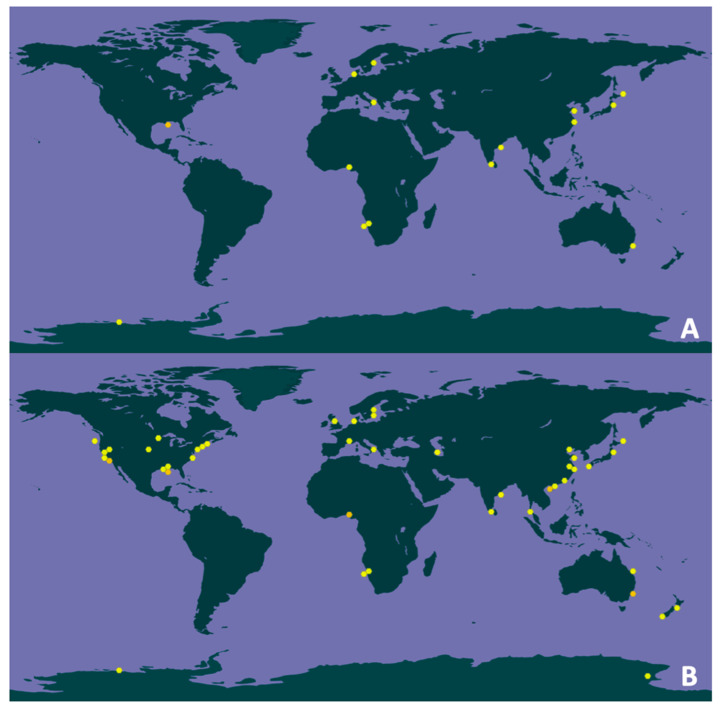
(**A**) Location of the 25 georeferenced records (among the total 35 occurrences) for the genus *Dissulfurirhabdus* in June 2020 based on the metagenomic 16S ribosomal RNA gene sequences from the GBIF database; (**B**) Location of the 114 georeferenced records (among the total 200 occurrences) for the genus *Dissulfurirhabdus* in June 2020 based on the metagenomic and metabarcoding 16S rRNA sequences from the GBIF database.

**Table 1 microorganisms-08-01132-t001:** General features and genome sequencing information for *Dissulfurirhabdus thermomarina* SH388T according to MIGS recommendations.

Item	Description
**Investigation**	
Strain	*Dissulfurirhabdus thermomarina* strain SH388^T^
Submitted to INSDC	GenBank
Investigation type	Bacteria
Project name	JAATWC000000000
Geographic location (latitude and longitude)	44°29.469′ N, 146°06.247′ E
Geographic location (country and/or sea, region)	Sea of Okhotsk, 250 m from the Kunashir Island shore (Sakhalin oblast, Russia)
Collection date	June 2013
Environment (biome)	marine hydrothermal vent biome ENVO:01000030
Environment (feature)	marine hydrothermal vent ENVO:01000122
Environment (material)	marine hydrothermal vent chimney ENVO:01000129
Depth	−12 m
**General features**	
Classification	Domain *Bacteria*
	Phylum *Proteobacteria*
	Class *Deltaproteobacteria*
	Not assigned to an Order
	Not assigned to a Family
	Genus *Dissulfurirhabdus*
	Species *Dissulfurirhabdus thermomarina*
Gram stain	Negative
Cell shape	short rods
Motility	Motile
Growth temperature	Thermophilic, optimum at 50 °C
Relationship to oxygen	Anaerobic
Trophic level	Chemolithoautotrophic
Biotic relationship	free-living
Isolation and growth conditions	DOI 10.1099/ijsem.0.001083
**Sequencing**	
Sequencing technology	Illumina MiSeq 2 × 150 bp
Sequencing platform	Molecular Research, MrDNA (Shallowater, TX, USA)
Assembler	Unicycler (version: 0.4.8-beta)
Contig number	36
*N50*	240,491
Genome coverage	1384.362×
Genome assembly NCBI	ASM1297923v1
Assembly level	Contigs
**Genomic features:**	
Genome size (bp)	2,461,642
GC content (mol%)	71.1
Protein coding genes	2267
Number of RNAs	50
tRNAs	47
16S-23S-5S rRNAs	1-1-1

**Table 2 microorganisms-08-01132-t002:** Occurrence of the thermophilic sulfur-disproportionating bacteria based on the GBIF (Global Biodiversity Information Facility) database.

Genus	Class	Occurrence		Georeferenced Records	
		Genus	Species	Genus	Species
*Dissulfurirhabdus*	*Deltaproteobacteria*	200	35 ^a^	114	25 ^a^
*Dissulfuribacter*	*Deltaproteobacteria*	230	27 ^b^	130	14 ^b^
*Caldimicrobium*	*Thermodesulfobacteria*	85	2 ^c^	39	1 ^c^
*Thermosulfurimonas*	*Thermodesulfobacteria*	27	17 ^d^	5	3 ^d^
*Thermosulfuriphilus*	*Thermodesulfobacteria*	0	NA	0	NA

The most abundant species of the genus are presented: ^a^
*Dissulfurirhabdus thermomarina*, ^b^
*Dissulfuribacter thermophilus*, ^c^
*Caldimicrobium thiodismutans*, ^d^
*Thermosulfurimonas dismutans* (Abbreviation: NA, not applicable).

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
