# Peer review of "Genomic Characterization and Environmental Distribution of a Thermophilic Anaerobe Dissulfurirhabdus thermomarina SH388T Involved in Disproportionation of Sulfur Compounds in Shallow Sea Hydrothermal Vents"

_microorganisms, 2020, doi:10.3390/microorganisms8081132_

Round 1

Reviewer 1 Report

The article by Allioux and co-workers is devoted to genomic characterization of a thermophilic anaerobe Dissulfurirhabdus thermomarina SH388T, which was earlier described by Slobodkina G., Slobodkin A. et al. (2016). The genus Dissulfurirhabdus was proposed based on the description of a single strain of a single species. Cultivation of this organism is difficult. Genome sequencing may be therefore used for better understanding of the metabolic potential of this new group of the Deltaproteobacteria. The authors consider thoroughly the genes of the central carbon metabolism, hydrogen and nitrogen metabolisms, and especially of sulfur metabolism. Comparison of the genes encoding the AprB proteins from different members of Thermodesulfobacteriales was carried out, as well as comparative analysis of the metabolic properties of bacteria disproportionating the sulfur compounds: Thermosulfurimonas dismutans, Thermosulfurimonas marina, and Thermosulfuriphilus ammonigenes. In the course of the genome analysis, it was revealed that this species could reduce sulfite but not sulfate, although the potential genomic resources were present. “Sulfur-disproportionating taxa do not necessarily have the same ecophysiological properties as sulfur-oxidizers and sulfate-reducers, and this could have a significant impact on our understanding of the biotic cycle of sulfur in natural environments.”

An independent work by Ward and co-authors (2020), which was published as a short communication several days before submission of the reviewed article, also provides the data on genome sequencing of this strains, albeit less complete than in the study by Allioux and co-workers. While Ward and co-authors (2020) proposed classification of Dissulfurirhabdus thermomarina SH388T within the family Dissulfurirhabdaceae fam. nov., they provided neither any data supporting this proposal, nor the formal description of the new family.

In order of discussion, the reviewer proposes the following. Since Microorganisms considers also the articles on prokaryote taxonomy, the reviewer's opinion is that Allioux and co-workers should have used this chance by adding a section dealing with taxonomy of Dissulfurirhabdus thermomarina SH388T. ANI and dDDH values could have been included into the article, provided they support classification of Dissulfurirhabdus and its closest relatives Dissulfuribacter and Dissulfurimicrobium within a new family. According to the results of genome sequencing of strain SH388T, its genomic G+C content is 71.1 mol%. This value is higher than 64.6 mol% given in the description of the species. Thus, it would have been advisable to provide an Emended description of this species and genus and to give a formal description of this family as well. The article title should have been changed accordingly.

The article by Allioux and co-workers is well-written. The work was carried out using up-to-date molecular and bioinformatic techniques, and may be published in the Microorganisms journal after minor revision.

There are, however, some comments. The authors should have paid more attention to formatting. Authors must use the Microsoft Word template or LaTeX template to prepare their manuscript. Citation rules accepted in this Journal should be followed. References must be numbered in order of appearance in the text (including citations in tables and legends) and listed individually at the end of the manuscript. Include the digital object identifier (DOI) for all references where available. The reviewer found a single correctly formatted reference. Line numbers should have been inserted throughout the manuscript.

Specific comments are listed below.

  1. 6, Table 1: General features and genome sequencing information for Dissulfurirhabdus thermomarina SH388T according to MIGS recommendations.

Table 1 as such is absent. The description below provides some genomic information according to MIGS recommendations. If no traditional table is planned in the text, the section heading shall be retained, while the wording “Table 1” should be excluded from the text.

References to Table S1 should be added in sections 3.2 and 3.3.

The title of Table 2 should be located above the table.

On Figures 2 A and B: Repeating areas should be removed from the maps: everything left of Alasks should be removed from the left part of Figures A and B, and everything right of the Bering Strait, from the right part.

Individual comments are marked in the text.

Author Response

Answer to reviewer 1:

First of all, we would like to thank Reviewer 1 for his detailed response and his positive comments and suggestions for enhancing the value of our work. In fact, our genome assembly had already been conscientiously annotated and analyzed when the article by Ward et al., 2020 was published. As our overall analysis is more detailed and complements their studies with additional material, we decided to submit our manuscript anyway and furthermore decided to write an additional section in the manuscript focusing on the AprB gene, to evaluate and discuss the hypothesis of Ward et al., 2020 associated with disproportionation.

Formal description of a new family

We thank the reviewer for his very interesting comments on this part. We have considered the reviewer's proposal to add a section on taxonomy and have carried out the suggested analyses. Unlike the species (and genus since the beginning of this year), there is currently no definition of a family of bacteria or archaea.

The only references found in the literature on this issue are the following:

- A paper by Yarza et al. (2014; Nature Reviews in Microbiology 12: 635-645) which indicates that the delineation threshold of the family is around 86.5% 16S rRNA gene sequence identity (Median sequence identity: 92.25%; minimum sequence identity: 87.65%)

- An article by Kim et al. (2014; Int J Syst Evol Microbiol. 2014;64(Pt 2):346-351) which states that “the ANI values calculated between strains belonging to the same family showed uneven distribution, having an obvious low-frequency area in the range of 81.0–96.0 % ANI and high frequency at >96 % ANI”

We calculated these proxies for the described strain and members of the new putative family as well as for the taxa closest to the new family. The results are not as clear as expected. Here is a summary of the results:

Taxa belonging to the putative new family Dissulfurirhabdaceae fam. nov.

Comparison to Dissulfurirhabdus thermomarina

Dissulfuribacter thermophilus

Dissulfurimicrobium hydrothermale

16S rRNA gene sequence identity (%)

91.6

90.4

OrthoANIu value (%)

66.66

(genome not sequenced)

Taxa just outside to the putative new family Dissulfurirhabdaceae fam. nov.

Comparison to Dissulfurirhabdus thermomarina

Desulfosoma profundi

Desulfacinum infernum

Desulfoglaeba alkanedexens

Desulfosoma caldarium

Desulfomonile limimaris

16S rRNA gene sequence identity (%)

88.3

88.18

87.82

87.73

87.72

OrthoANIu value (%)

(genome not sequenced)

67.54

67.78

66.09

(genome not sequenced)

The taxonomic description of the new family would have added real value to the paper, and was an excellent suggestion, but in our opinion, the definition of the microbial family is not clear and the results obtained here are not robust enough to propose a new family simply on the basis of these data and the published physiological data on these strains. Consequently, we have not added a formal description of the family in this article.

Genomic G+C content

The DNA G+C content value was initially determined from the melting point with DNA of Escherichia coli K-12 as a reference. This method is more biased than determining the percentage of G+C directly from the genomic sequence. We have therefore proposed to amend the description of the species Dissulfurirhabdus thermomarina as follows:

Emended description of Dissulfurirhabdus thermomarina (Slobodkina et al., 2016): The description is as given by Slobodkina et al. [7], with the following amendment: The genomic DNA G+C content of the type strain is 71.1 mol%.

Article formatting

We have taken into account all the reviewer's remarks. Citations have been modified according to the journal format and numbered. The line numbering and Table 1 that were present on the initial version submitted probably disappeared during the automatic formatting during submission. We have added them again. References to Table S1 have been added in sections 3.2 and 3.3 but also 3.4 and 3.5. The title of Table 1 and 2 have been moved above each table. Figures 2 A and B have been reframed as suggested. Individual comments marked in the text have been applied or modifications have been done.

Reviewer 2 Report

The manuscript analyzes the draft genome sequence of Dissulfurirhabdus thermomarina SH388. The analysis is very thorough and explained well.  My primary concern with the manuscript is that the genome of this organism was just published by another group (Ward et al., 2020).  I'm guessing the genome data are largely the same.  The analyses in the Allioux et al. manuscript are far more thorough than in the other manuscript; however, I question the value of having two genome sequences for the same organism.  This point is moot because both sequences have already been deposited into genome sequence databases.  Therefore, it will be up to interested researchers to choose which sequence data and analysis they will use.

Author Response

Answer to reviewer 2:

We would like to thank Reviewer 2 for his response and his positive remarks. In fact, our genome assembly had already been conscientiously annotated and analyzed when the article by Ward et al., 2020 was published. As our overall analysis is more detailed and complements their studies with additional material, we decided to submit our manuscript anyway and furthermore decided to write an additional section in the manuscript focusing on the AprB gene, to evaluate and discuss the hypothesis of Ward et al., 2020 associated with disproportionation.

In addition, as the reviewer said, two different versions of the genome are now available which may be useful from different perspectives and will leave the choice to future users.